# Effects of Miniaturization on the Quality of Metallized Plastic Parts

**DOI:** 10.3390/mi13040515

**Published:** 2022-03-26

**Authors:** Aminul Islam, Hans Nørgaard Hansen

**Affiliations:** Department of Mechanical Engineering, Technical University of Denmark, 2800 Kgs. Lyngby, Denmark; hnha@mek.dtu.dk

**Keywords:** miniaturization, metallization of plastics, injection molding

## Abstract

The metallization of plastics is an important industrial process. Plastics are metallized for both aesthetic and functional purposes. The unceasing pursuit towards the miniaturization and reduction in the part’s size challenges the already complicated process of metallization. A rigorous research study uncovering the effects of miniaturization on the quality of metallized parts is missing at the state-of-the-art level. This study focuses on the quality of the deposited metal film based on geometrical dimensions and systematically characterizes the effects of miniaturization on the metallized micro-components. The experimental results presented in this paper reveal the hidden synergy among the metallization quality, part dimension, and process conditions used both for substrate fabrication and for metallization. The paper broadens the fundamental understanding about the interactions of various design, materials, and process parameters involved in the manufacturing process chain. The results and discussions presented in this paper will be valuable sources of information to deal with the integration of micrometallic structures on polymeric substrates for high precision applications.

## 1. Introduction

Metallization is the general name for the techniques of coating metals on the surface of any objects [1]. Metallic coatings can be decorative, protective, or functional and used in huge varieties of applications such as electronics [2], Molded Interconnect Devices (MIDs) [3], automotive, aerospace, plumbing, packaging [4], Micro Electro Mechanical Systems (MEMS) [5], etc. The modern age is the age of smart devices, where technical systems are becoming complex with the integration of advanced mechanics, sensors, and electronics. With the advent of the concepts such as Cyber-Physical Systems, Internet of Things, Big Data, Personalized Healthcare, etc., the integration of metallic structures on polymer substrates will become even more important for the industries [3]. The ever-increasing demand for miniaturization across industrial sectors possesses big challenges for metallization and selective metallization processes. As the further miniaturization and reduction in the part’s size are a technological reality [6], it is important to know the effects of miniaturization on the quality of metallization. Almost nothing is reported on this issue at the state-of-the-art. The experimental investigations presented in the paper systematically characterize the effects of miniaturization of the plastic components on the quality of the achieved metallized parts by an electroless process. The results and the discussions presented in the paper form a valuable source of information for researchers from both industry and academia with respect to the integrations of metallic structures on polymer substrates for highly demanding applications.

## 2. Electroless Metallization of Plastics and State-of-the-Arts

The metallization process has been introduced at the commercial level to the industry in the early 1960s. Covering plastics by metals allows the combination of properties of both material groups and, therefore, extends their applications by making hybrid and functionally versatile components [3]. Lightweight, corrosion resistant, flexible, and easy to process plastics gain electrical conductivity, luster, wear resistance, electromagnetic shielding, weatherproofing, etc., by the application of metallic coatings [4]. The metal layer on the plastic surface can be deposited by using different methods such as plating, vapor deposition, painting, spraying, printing, hot stamping, and so on. However, the most common method to metallize plastics currently is electroless plating as it is suitable for large-scale production [4]. Electroless metallization is also cheaper as no expensive equipment or high temperature is needed [7]. The electroless process is solely driven by a chemical reaction and does not use any external power supply; hence, the process is called the electroless process. Here, the deposition of a metal is performed by an autocatalytic process that involves a flow of electric charges (electrons) from the less electronegative metal or a reducing substance contained in a plating bath to metal ions being deposited [8]. The electroless process builds metal layers on the substrate surface in an atom-by-atom fashion. Depending on the application, metals such as copper, silver, gold, and palladium are used. The basic principles of the electroless metallization scheme and the experimental metallization setup are shown in Figure 1. 

Before the plating, the surface of the plastic material needs to be prepared in several steps [9]. An etching process, usually performed by oxidizing solutions (most often chrome acid), causes the surface to become hydrophilic through the generation of polar groups (carbonyl, carboxyl, and sulfate) and is roughened by the formation of micrgrooves. The microgrooves produced on the plastic surface ensure good bonding between the plastic and the deposited metal during the metallization process [9,10,11]. After etching, the surface is neutralized by removing residues of the etchant. The neutralization of the surface is usually performed by dipping the parts in hydroxylamine hydrochloride (NH_2_OH.HCl) for 5 min [9,11]. The neutralized surface becomes ready for an activation process. Activation agents are usually colloidal solutions based on precious metals such as palladium, platinum, or gold surrounded by tin ions (Sn^2+^). Due to the activation process, Sn^2+^ can wet the surface of the substrate and presents a “sensitized” surface for metal deposition [9]. In addition, the small particles of the noble elements such as palladium are deposited in micropores that are formed during the etching process and play a role as an active catalyst for electroless plating. At the beginning, the copper atoms are deposited on the catalyzed surface and then merged with each other to form a continuous metal layer [9,11]. Baths used for metal deposition contains metal ions, stabilizers, and reducing agents. In general, the electroless process enables the deposition of a very homogenous coating, even on structures characterized by superfine dimensions or complex geometry [12]. The transition from macro to micro can influence the mechanisms that are responsible for deposition and adhesion between low surface energy plastic and high surface energy metallic material. An important question to ask is the following: What happens to the quality of the metallized parts when the part or feature dimensions are gradually decreased and where is the limit for metallization? The state-of-the-art literature cannot provide scientifically proven or experimentally validated answers to this question. The quality of metallization is dictated by operating conditions (temperature, pH, and agitation), the composition of the metal bath, and by surface topography [13]. Problems such as metal electromigration, corrosion, and diffusion are expected to compound as metal feature sizes continue to shrink [14,15]. These issues may ultimately limit the ability of the features to function as useful electrical interconnects. The deposition rate and the adhesion of the deposited metal may also be affected by decreasing dimensions and low volume of the bulk material to be metallized [2]. For selective metallization, e.g., in the case of MIDs, a special challenge is to achieve a sharp and well-defined line (having a clear boundary and sharp line features easily distinguishable from the substrate surface) in microscale between metallized and non-metallized areas [16]. Another challenge with microscale metallization is through-hole metallization (metallizing the inner wall of holes with diameters in the micro range) [17]. An important consideration for metallization of microparts is the surface-to-volume ratio. The surface-to-volume ratio increases with decreasing part dimensions [18]. This affects surface-to-surface interactions, heat transfer properties, wetting properties, and collections of contaminations. For the parts with smaller dimensions, the effect of contaminations and flow properties of metallization fluids can be more critical than on the parts with relatively larger dimensions. The metallization quality of plastic parts can be influenced by the production process of the plastic part. Molding of the micro part is a challenging process and the quality of the molded microparts will, with no doubt, affect the quality of the metallized part. The scope of the following investigation is to find the effects of the omnipresent trend of miniaturization on the quality of the metallized parts.

## 3. Experimental Investigation

The main objective of the experiment conducted for this investigation was to systematically characterize the effects of the decreasing feature dimension on metal film deposited using the electroless plating process. In addition, some general concerns with the electroless plating of microparts, especially for critical areas in the parts such as internal and external corners, edges, and fingertips, are addressed with the investigation. The discussions below focus on the materials, test part, and methods used to fulfill the objectives together with the analysis of the obtained results. 

### 3.1. Materials and Methods

Two thermoplastic materials were selected for manufacturing plastic parts. One was acrylonitrile butadiene styrene (ABS), while the other one was a blend of polycarbonate (PC) with acrylonitrile butadiene styrene (ABS) characterized by better mechanical properties (higher tensile strength and Young’s Modulus comapared with pure ABS) compared with the pure ABS material. ABS and PC-ABS blend are the most commonly used materials for plating applications. ABS was of commercially available grade named Terluran GP-35 from INEOS Styrolution, Germany. The blend material was Bayblend T65 PG from Covestro Deutschland AG, Germany. Both materials were characterized by high flowability and good mechanical properties and are suitable for electroless plating. These made them good candidates for the investigation. For the metallization of plastic microparts, copper was selected as this is the most widely used material for electroless plating of plastics in industrial applications. As the test part, a model with decreasing feature dimensions was developed. The CAD design of the part and its detailed technical drawing are shown in Figure 2a,b. The part consisted of a 0.2 mm thick semicylinder and eight ‘fingers’ with different heights and widths on the opposite side. Finger heights were determined by the outer perimeter of the circle defining the cylindrical part, while widths of the fingers decreased from 0.22 to 0.14 mm with a constant spacing of 0.75 mm between two neighboring fingers. The dimensions of the fingers were selected based on the commonly used ranges of micrometallized parts and what was suitable for the injection molding process. To carry out a consistent evaluation of manufacturing results in the further part of the investigation, the fingers were numbered from F1 to F8, as presented in the Figure 2a.

Injection molding was chosen to produce the part with both selected materials. The process was carried out using an Engel Victory Tech 80/45 molding machine (Figure 2c). A single-cavity mold was employed, and for the molding parameters, the materials suppliers’ recommendation was followed. Metallization process steps that were applied to produce copper film on plastic parts are shown in Figure 1b. The reagents and conditions applied for all stages are presented in Table 1. Deionized water was used for rinsing after each step. For Cu deposition, the bath Circuposit 3350 from Shipley Corporation was employed with a magnetic stirring system at 200 rpm.

### 3.2. Characterizations

The quality of the electroless copper plating was evaluated with the light optical microscope (LOM) Zeiss Stemi 2000-C and scanning electron microscope (SEM) JSM-5900. For imaging, continuous zooms from 8× to 40× were applied for LOM, and for SEM analysis, variable zooms from 300× up to 3000× were applied to explore the structure of the metal coating in detail. The focus was placed to find the connection of metallization quality with the decreasing feature size of plastic parts. Moreover, the metallization quality of microfeatures in critical areas was thoroughly investigated. In order to investigate cross-sections of microfeatures, a special sample preparation was required. Initially, the samples were embedded in silicon epoxy and then grinded and polished according to Struers’ recommendations [19]. Before placing the samples into the SEM chamber, a thin carbon film was deposited on their surfaces to avoid the accumulation of static electric charges, which could limit image information. With the help of epoxy molded samples -, the thickness of deposited coatings was examined together with the continuity and uniformity of the coatings around the microfeatures. The thickness was measured from SEM images. For each finger, the measurement was taken in ten different locations; finally, the average was taken and standard deviation was calculated. To assess the surface roughness of the samples before and after electroless plating, the Sa parameter (arithmetical mean height) was taken into account following the recommendation of EN ISO 25178 [20]. The measurement was conducted by Alicona Infinite Focus microscope (model IFMG4g). The software package Scanning Probe Image Processor (SPIP) issued by ImageMetrology was used to compute roughness parameters, and global levelling was applied in the SPIP to process images as accurately as possible.

### 3.3. Results and Analysis

The LOM analysis of metallized microparts molded from ABS shows that the electroless copper plating allows covering the surface with relatively uniform and homogenous copper films without any visible metal accretions or deficiencies (see Figure 3). The quality of the coating seems to be better on the large flat area of the part (Figure 3a), although the fingers are also well-covered (Figure 3b). The deposited layer has the characteristic metallic color of copper. There is no visible difference in the coating quality deposited on fingers with different dimensions. The finger with the smallest width is completely covered, as well as the finger with the largest width. It is observed that all ABS parts are uniformly covered by the metal, including the irregular flashes and their sharp edges, and peaks formed during the microinjection molding process. Despite the fact that the fingertips are characterized by the lowest surface areas, copper was able to create continuous and defect-free coatings (Figure 3c). This means that the small surface area of ABS is capable of reacting properly with applied etching and activating solutions to provide a surface that is ready for subsequent metallization at the same level as large flat area of the part. The surface area and its topography are important for metallization. For successful metallization, sufficient interactions among the surface and metallization chemicals are needed so that the necessary changes of the surface can be made for metallization. For the experimental test part, the fingertips having the smallest surfaces could be critical for metallization. However, the result shows that metallization on the fingertips was possible even at the smallest finger, and they were not critical for metallization, at least for the dimensional range chosen for the experiment. Therefore, with this investigation, the requirement for minimum surface area for electroless metallization of polymers was not experimentally proven, and this provides an area for future investigations. 

Figure 3d shows effective copper depositions on internal corners, where fingers meet the semicylinder. The quality of metallization in this area seems a bit lower where some uncoated black areas are visible. The reason for this is that access for copper ions from the deposition bath could be impeded due to obstructions to the fluid flow created by the fingers. All sharp edges presented in the Figure 3d–f are also well-coated. Nevertheless, there are some film discontinuities (low deposition-black areas) appearing close to the gate system of the part (Figure 2f). This is the result of the mechanical damage of the plastic part during the molding process associated with the high shear of the material in the gate area.

The quality of the metallization of PC-ABS specimens (see Figure 4) is noticeably worse than the quality of the ABS parts. The copper coating covering the surface is non-uniform, and this is apparent not only on the fingers but also on the semicylinder area (Figure 4a). The fingertip surface around the semicylinder edges and around the fingers have poor metallization quality (Figure 4c–e). The film is discontinuous on the sharp edge. Compared to the metallization quality at the same spots on ABS parts, a significant difference can be observed. There are many uncoated spots on the micropart fingers and on the fingertips. Since the film’s quality is equally poor on the smallest and the largest areas of the blend material compared with the similar spots of the ABS material, it suggests that there is some intrinsic issue with the polymer material that is causing the difference.

What is interesting for the pictures presented in Figure 4 is a pattern of film discontinuities diverging radially from the end of the gate system. It is particularly visible in Figure 4f. The pattern reflects a direction of the melt flow from the gate into the mold’s cavity. The characteristic appearance of metal film is a result from previous step in the process chain-injection molding. Due to the high viscosity of the PC-ABS blend and small size of the micromolding gate, the melt was exposed to high shear forces during the molding process. Near the mold wall, the shear rate is higher due to the fountain flow effect of polymer. This provided much larger deformations for the material in contact with the mold than the material in the core, which resulted in the formation of the characteristic ‘skin-core microstructure’ (Figure 4b) [22]. In case of the PC-ABS blend, ABS is dispersed into the PC matrix. SAN domains (styrene-acrylonitrile resin) from ABS are elongated on the micropart’s surface and polybutadiene rubber particles are randomly grafted (black dots in Figure 4b). However, at the surface and within a few micrometer depth from the surface, due to the extremely high shear rate, SAN lamellae become free of rubber particles, leaving the SAN-grafted rubber particles dispersed below in the polycarbonate phase. Thus, the part surface is made mainly by deformed SAN lamellae free of rubber particles. Due to the higher resistance of SAN to the etching solution [22], some regions of the surface remain uncovered during the metallization process, producing the characteristic appearance of copper coatings.

The high-resolution analysis performed using SEM confirms the results obtained from the LOM analysis that the copper film deposited on ABS had much better surface quality than the film on PC-ABS. Figure 5 presents a comparison between the microstructure of copper films deposited on different plastic substrates. It can be observed that copper atoms are well distributed over the ABS surface, forming a compact coating with limited porosity (Figure 5a). On the other hand, the morphology of copper film on the blend is not so well developed as in the case of ABS (Figure 5b). Copper particles are more irregular compared to grains deposited on ABS. The uncovered areas of PC-ABS reveal insufficient catalytic action on the surface due to resistance to the etching chemical; thereby, copper deposition was unsuccessful in many areas.

#### 3.3.1. Metallization Quality on Different Microfeatures

Figure 6 presents a comparison between coating qualities on both plastics in the same areas that were suspected to be the critical for metallization. Figure 6a,b show how well the copper film is deposited on fingertips, which are characterized by the smallest surface areas. Figure 6c–f present the quality on finger walls and edges, while Figure 6g,h show the comparative quality of metallization at the base area where the finger meets the semicylinder part. An assessment of metallization effectiveness on internal corners can be performed by comparing Figure 6g,h. Similarly to LOM analysis, these high resolution images also confirm that electroless copper plating provides better metal finishes on ABS microparts for all investigated areas (top surfaces, side walls, external edges, and internal corners).

Coatings on the ABS materials in the critical areas are compact, continuous, and smooth, with less visible defects compared to the coating on PC-ABS. However, here, it can be noticed that the copper coating on ABS is not as smooth as it seemed during the LOM observations. There are some porosities on side walls (Figure 6e) and top surfaces of fingers (Figure 6a); nevertheless, the indicated imperfections are significantly smaller compared to the same spots of the blended substrates (Figure 6b,f, respectively). In the case of the blend, metallization has been noticeably less effective on the edges and fingertips, where large areas of uncovered plastic are evident.

#### 3.3.2. Metallization Quality on Different Fingers

Figure 7 shows the results of copper plating on the surfaces of fingers, which differ by materials and dimensions. The fingers shown in Figure 7a,b are molded from ABS, and the two other fingers shown in Figure 7d,e are from PC-ABS. Figure 7a,d are taken from the tip of narrowest finger (F8) and Figure 7b,e are from the tip of widest finger (F1). There is again a noticeable difference in the quality of the copper coating based on the two different plastic materials. The copper film covers ABS fingertips completely, while on the blend, complete coverage is missing. An important thing to observe here that is the change of dimensions for the microfeatures does not have a noticeable influence on metallization quality. The same observation was also made with the other fingers. Figure 7c,f present representative SEM pictures of the copper coating around the fingers of both plastics at high magnification (the pictures were made on the cross sections of F8). It is clearly observed that the film covering ABS (Figure 7c) is more continuous and stable in terms of the thickness, compared to the PC-ABS part. Metal deposits irregularly on PC-ABS, forming a coating discontinuous in many places (Figure 7f).

Figure 8 helps to qualitatively evaluate the effectiveness of electroless plating around different fingers at different finger heights for both materials. The cross sections of the fingers were prepared in two different finger heights: ‘close to the base’ cross section was about 0.5 mm from the base and another cross section ‘far from the base’ (close to the fingertip) was prepared at 1.5 mm distance from the base of the fingers. Figure 8a shows the example of the finger cross sections prepared with the help of the epoxy mold for this investigation. SEM pictures presented in Figure 8b–e are taken from ABS parts and Figure 8f–i from PC-ABS part. Figure 8b,c,f,g are taken from the widest finger (F1) of the ABS and PC-ABS parts and Figure 8d,e,h,i are taken from the narrowest finger (F8) of the ABS and PC-ABS parts. The first difference, which is easy to notice, is the diverse shapes of finger cross sections. According to the technical drawing, they should be the same and the finger cross section should be rectangular in shape. The different widths of the fingers presented different degrees of filling difficulties and caused variations in finger shapes. Furthermore, different morphologies of plastics can be observed by comparing both sets of pictures. Pictures in Figure 8b–e indicate that the material inside metal frame is homogeneous, while Figure 8f–i show that clear multiphase microstructure could be identified. 

The important fact is that the decreasing size of the finger did not affect the quality of electroless copper plating. Moreover, the copper coating looks better on the finger with smaller size, particularly for ABS samples (Figure 8d,e). It is thicker and discontinuous in a lower number of places compared to the larger fingers (Figure 8b,c). The same conclusions can also be made for the finger cross sections of the blend material (Figure 8f–i). This indicates that scaling-down the part dimensions and the bulk volume of the plastic material did not negatively affect metallization quality. It is clear from the investigation that F8 with a width of 140 µm is not the limit for electroless plating; features sizes smaller than this can also be metallized. The coating on ABS finger looks better on samples, where cross sections were made ‘far from the base’ of fingers (Figure 8c,e). The film is thick and continuous on the entire finger perimeter; however, its thickness varies quite significantly. The quality of the copper deposit close to the base is noticeably worse for both fingers (Figure 8b,d,f,h).

#### 3.3.3. Film Thickness on Different Fingers

Figure 9a presents results of film thickness measurements for different materials, fingers, and measured positions. It is quantitatively proven that the copper coating on ABS is thicker than on PC-ABS for all measured fingers. The thickness of the copper coating on ABS varies between 2.5 and 3.5 µm; while on the blend, it varies between 1.5 and 2.5 µm. High values of standard deviations, as observed with thickness values in Figure 9a, result from irregular topography of the copper deposit around fingers for both plastics. The film thickness is higher for ABS parts, which means that the copper is capable of depositing more atoms on ABS than on PC-ABS in the same time period, indicating that ABS is a more suitable plastic for the electroless plating process. A relation is observed indicating the influence of the finger size on metallization quality. For both materials, F1 (widest finger) is characterized by the lowest thickness values, compared to the other two fingers with smaller widths. Moreover, the film’s thickness on F8 (narrowest finger) is higher than other fingers. This means that the decreasing dimensions of the microfeature do not have negative influences; rather, it shows some positive effects on the thickness of the deposited coating. The finger with smaller width allows obtaining thicker coatings with higher quality compared to larger fingers. The deposition rate is equal in the entire deposition bath; nevertheless, it seems that copper builds up faster around the smaller fingers smaller overall volume of materials. Furthermore, a relation between the film thickness and the finger height is identified for both plastic materials with current investigations. In all cases, copper thickness is higher on the sections close to the fingertip than close to the base of the fingers. This results from having easier access to metallization chemicals around the fingertip areas than around the bottom part of the fingers. The conclusions made with the quantitative analysis coincides with the previous qualitative assessment for the continuity and uniformity of the copper coating.

#### 3.3.4. Roughness Measurement

The roughness measurement results are presented in Figure 9b. It shows that both the untreated ABS and the PC-ABS blends are characterized by relatively low surface roughness (Sa), where blend materials have slightly higher roughness (mean values of 471 nm and 501 nm, respectively). After the etching step, the value of Sa increases for both materials, keeping the previous trend (629 nm for ABS and 655 nm for PC-ABS). Activation and subsequent reduction stages do not have a significant effect on the roughness for the blend, whereas for ABS, they make the surface slightly smoother. Eventually, just before metal deposition, the surface of ABS is finer than the surface of the blend (549 nm and 672 nm, respectively). This higher roughness for the blend material contributed to slightly higher adhesion of the copper on the plastic substrates, which was confirmed qualitatively when performing a cross-cut test according to the standard DIN EN ISO 2409 [23]. The measurements performed on microparts after metal deposition show that the copper film on ABS has slightly better surface finishes (491 nm) compared with PC-ABS (536 nm).

## 4. Discussion

It was experimentally proven that ABS is a more suitable plastic than the blend for electroless plating. The experimental results clearly show that the finger with the smaller width allows obtaining a thicker coating with higher metallization quality compared to the larger fingers. Copper builds up faster around the smaller fingers even when the deposition rate is equal in the entire deposition bath. Thus, the question is as follows: What causes smaller fingers to get better metallization? This is not experimentally proven, but one factor could be the surface-to-volume ratio of the fingers. Smaller fingers have larger surface-to-volume ratios compared with the larger fingers; thus, the catalyzed palladium sites can appear in larger numbers for smaller fingers and can be placed closer to each other. This may happen simply because of the smaller dimension of the part and higher surface-to-surface interactions of the palladium particles with relatively larger surfaces provided by the smaller fingers (due to their high surface to volume ratio). Therefore, when metallization starts on the deposited palladium sites, the merging of metallized areas proceeds faster, resulting in more continuous and higher quality metal films. One fundamental step for metallization is the etching step. During this process, the surface roughness of the substrate surface is changed with the use of the etching chemical, and this can also be affected by the surface-to-volume ratio of the parts or features. Rougher surfaces can be made, and a better quality can be achieved for metallization as seed metal particles such as Pd can be better attached on the rougher surface during the activation process and can initiate improved metallization. This change in the surface roughness during the etching process will be larger if a larger surface in relation to the volume of the part is provided. For a larger surface, the surface-to-surface interactions between the substrate and chemicals are higher. This means that the degree of change of the surface’s roughness can be altered by etching process and is influenced by the surface-to-volume ratio. Moreover, the heat transfer rate and van der Waals attraction force (higher for the larger surfaces) and overall surface chemistry are different for larger surfaces provided by the parts with high surface-to-volume ratio. All these parameters connected to the surface-to-volume ratio of the part can make a positive difference in metallization quality.

Experimental results also proved that the easy access of metallization solutions to the surface to be metallized was important, and this was the reason why the metallization quality was better close to the fingertips. Electroless copper plating at the micro-scale was possible for both plastic materials used in the experiment, although the coating quality differed significantly. The comparatively worse plating results for PC-ABS resulted from high shear stress from the molding process, which contributed to the formation of characteristic ‘skin-core microstructure’ affecting the subsequent metallization process. The results showed that the metallization quality on the different cross sections of a micro finger was not the same. There were no build-ups of metals on sharp corners, edges, or on the fingertips. It was proven that metallization quality is highly influenced by the quality of the molded parts. In addition to the dimensional and physical quality of the molded parts, the surface and morphological properties generated during the molding process greatly affect the quality of the metallized parts. With the investigated micro range, the minimum feature size for metallization could not be specified, since the coating quality on the smallest finger was undeteriorated compared to the quality received on wider fingers.

## 5. Conclusions

The dynamic development of manufacturing technologies for electrical and electronic systems, automation, robotics, automotive, interconnect technologies, etc., triggers the need for the further development of metallization processes. The industry is required to produce metallic tracks and surfaces with high deposition quality in order to ensure high electrical, mechanical, and aesthetic properties of the metallized parts. The current research study was initiated to address the general concern of the decreasing dimensions of the plastic parts and its effects on the quality of metallized parts. The main focus of the research was placed on the investigation of the quality of deposited film on the plastic substrates in the microscale. The outcome of the research brings the confidence that metallization is not the critical process in related process chains; the critical process is the production of the substrates—most commonly, it is the injection molding process. Producing components with high aspect ratios or microdimensional parts by the injection molding process is always a challenge [24]. However, the current work reveals that molding can produce metallizable materials, and the electroless process can metallize these materials. Metallization quality is highly affected by the quality of the molded part. For the best metallization quality, the process parameters for both steps (micromolding and metallization) need to be carefully optimized to obtain the best quality for the metallized parts.

One important aspect of the metallization of plastic parts is the adhesion between the metal and plastic surface. This aspect was kept out of the scope of the current paper. Future investigations should be carried out to find the effects of miniaturization on the adhesion of the deposited metal on the part’s surface. With the current investigation, the requirement for minimum surface area for electroless metallization was not experimentally proven. This also provides a scope for future investigation to see if there is a lower limit for the surface area to be metallized by electroless processes. It seems from the current investigation that there is a correlation between metallization quality and surface-to-volume ratio of the part. This correlation is analytically discussed in the paper but no experiment was carried out to establish this analogy. Future investigations should be conducted to present experimental evidence for this correlation and to see if there is any causation between the quality of metallization and the surface-to-volume ratio of the parts.

## Figures and Tables

**Figure 1 micromachines-13-00515-f001:**
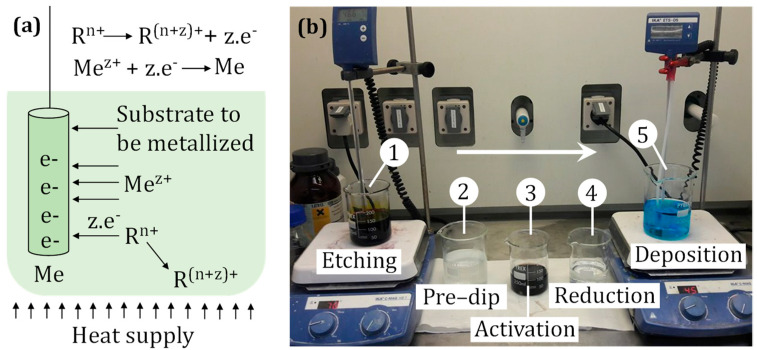
Basic principle of electroless plating—electrons from the surface reduce metal ions from the electrolyte leading to the metal deposition (**a**); the setup used for metallization during the current investigation (**b**).

**Figure 2 micromachines-13-00515-f002:**
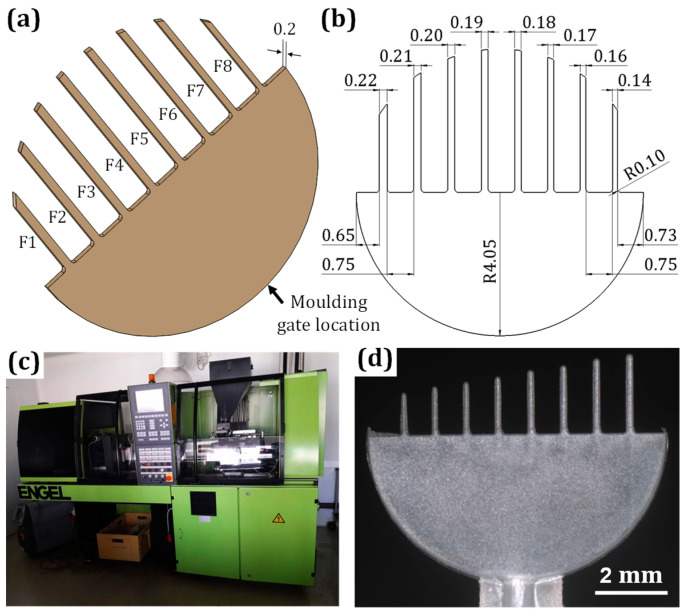
CAD model (**a**) and 2D drawing (**b**) of the chosen test part; molding machine used for the production of plastic parts (**c**); and a molded plastic part with ABS material (**d**).

**Figure 3 micromachines-13-00515-f003:**
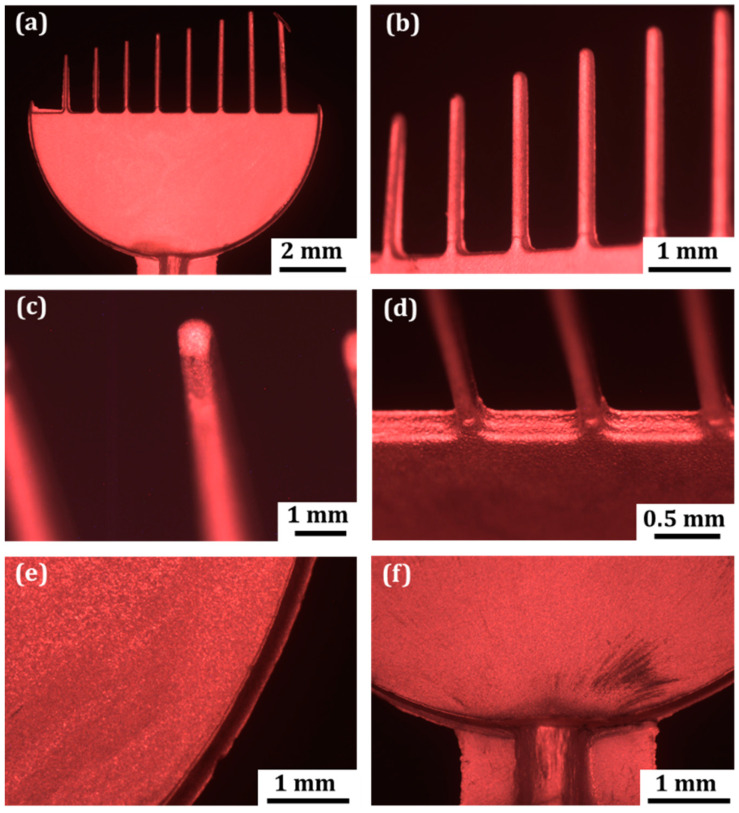
Quality of metallization on ABS micropart and on various features of the part: on the large semicylinder area (**a**); on the fingers (**b**); on the fingertips (**c**); on the internal corners (**d**); on semicylinder edge (**e**); and on the gate system (**f**).

**Figure 4 micromachines-13-00515-f004:**
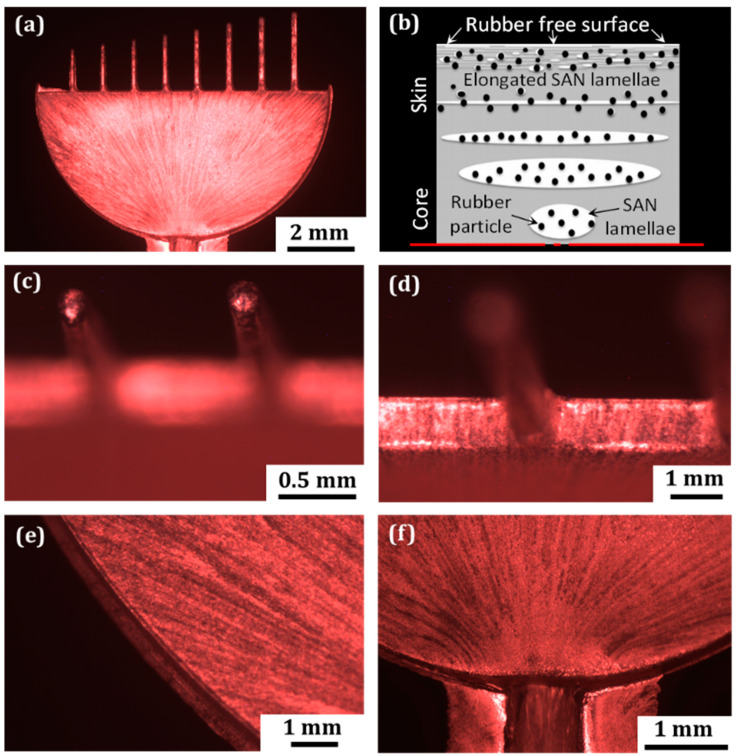
Poor quality of metallization on PC-ABS micropart (**a**); morphology of molded PC-ABS negatively affecting the metallization quality ((**b**) edited from [21]); low effectiveness of the metallization on PC-ABS microfeatures: on the smallest fingertips (**c**); on the internal corners (**d**); on semicylinder edge (**e**); and on the gate system (**f**).

**Figure 5 micromachines-13-00515-f005:**
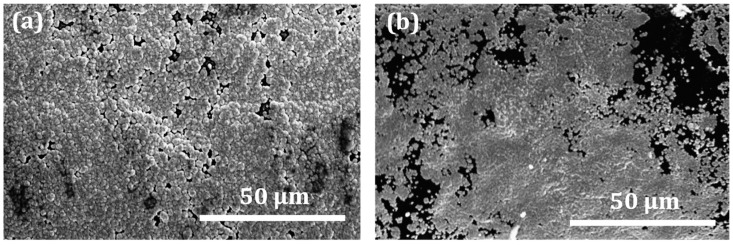
Morphology and microstructure of the copper coating on ABS part (**a**) and PC-ABS part (**b**).

**Figure 6 micromachines-13-00515-f006:**
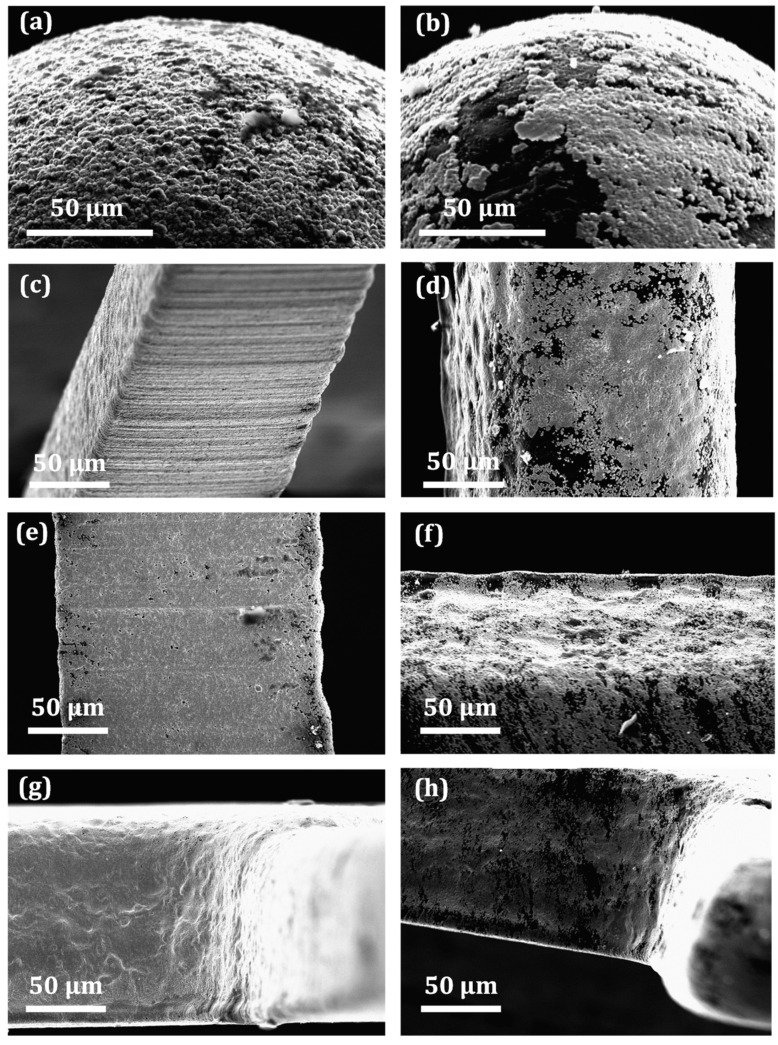
Quality of metallization on critical areas of the part—better metallization on ABS (left column), worse on PC-ABS (right column): fingertips (**a**,**b**); side walls (**c**,**d**); edges (**e**,**f**); and corners (**g**,**h**).

**Figure 7 micromachines-13-00515-f007:**
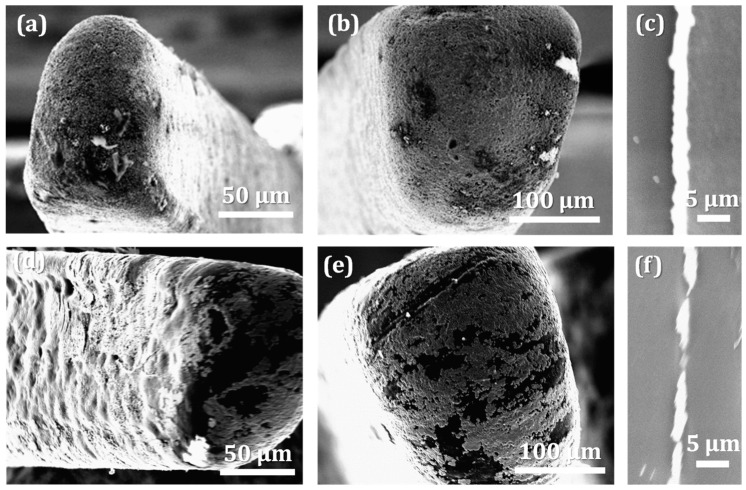
Metallization quality at the fingertips: ABS F8 (**a**); ABS F1 (**b**); PC-ABS F8 (**d**); PC-ABS F1 (**e**). The cross section of copper coating on ABS F8 (**c**) and on PC-ABS F8 (**f**).

**Figure 8 micromachines-13-00515-f008:**
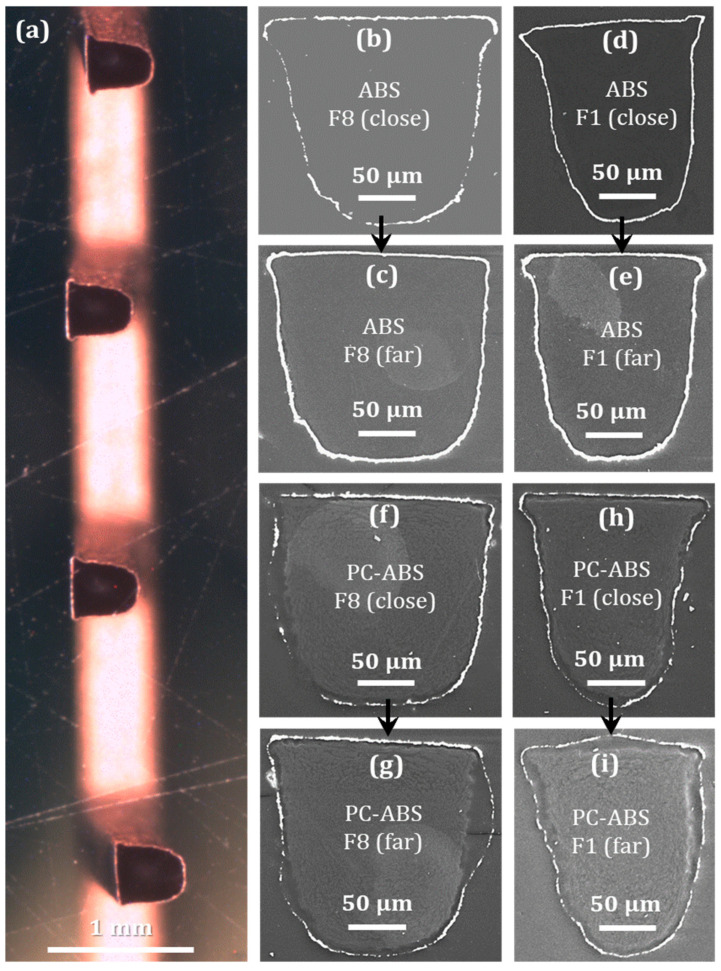
Epoxy molded and grinded sections of micro fingers (**a**). Quality and thickness of Cu coatings on ABS part (**b**–**e**) and PC-ABS part (**f**–**i**) at different fingers and heights (close to the base and far from the base).

**Figure 9 micromachines-13-00515-f009:**
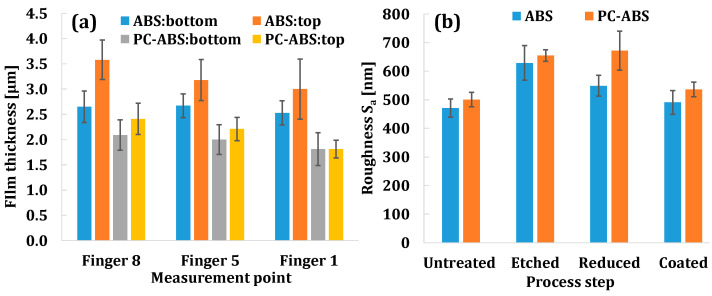
Thickness of the copper film on different fingers and finger heights (**a**); comparison of the surface roughness of plastic substrates after different preparation steps (**b**).

**Table 1 micromachines-13-00515-t001:** Agents used for electroless copper deposition [9].

Step	Composition	Temp.	Time
Etching	375 g/L CrO_3_ + 400 g/L H_2_SO_4_ (conc.)	63 °C	10 min
Pre-dip	2 g/L SnCl_2_ 2 H_2_O + 180 g/L NaCl + 100 mL/L HCl (conc.)	Room temp.	2 min
Activation	180 g/L NaCl + 120 mL/L HCl (conc.) + 1 g/L SnCl_2_ ∙ 2 H_2_O + 30 mL/L Pd catalyst	Room temp.	3 min
Reduction	300 mL/L HCl (conc.)	Room temp.	10 s
Deposition	Shipley Circuposit 3350	45 °C	10 min

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
