# Peer review of "Effects of Miniaturization on the Quality of Metallized Plastic Parts"

_micromachines, 2022, doi:10.3390/mi13040515_

Round 1

Reviewer 1 Report

The metallization of plastics is an important industrial process. The authors are aware of this and have submitted very important studies for review. Plastics are metallized for both aesthetic and functional purposes. Constant pursuit of miniaturization and reduction. Resizing parts is a challenge for an already complex metallization process. This work focuses on the quality of the deposited metal foil based on geometric dimensions and systematically characterizes the influence of miniaturization on metallized micro-components. The experimental results presented in the article reveal a hidden synergy between the quality of metallization, the dimensions of the parts and the process conditions used both for the production of the substrate and for metallization. The article enhances the basic understanding of the interactions of the various designs, materials, and process parameters involved in the manufacturing process chain.

The results and discussion presented in the article will be valuable sources of information on the integration of micrometallic structures on polymer substrates for high precision applications.

However, the article requires some corrections:

  1. The article needs to be sorted out. For example Chapter 3.2 is a problem . A correction needs to be made in this respect.
  2. In my opinion, in line 70 it should be mentioned what the surface is neutralized with.
  3. In lines 70 to 80 there are no technological details defining the described processes. Supplementing this fragment will allow to use other literature data and complete the references.
  4. The database of references should be expanded.
  5. In line 114-117 it is written about the better properties of one of the materials, this should be detailed.
  6. I also propose to consistently use the marks F1 - F8 in the description (description in the text other fingers, line 270-283)

It is worth noting that the research presented by the authors was described in great detail. An interesting thread would be to join them for adhesion or abrasion tests. But maybe in another job.

Author Response

Please find out point-by-point response to the reviewer’s comments in the attached file. 

Reviewer 2 Report

The authors are discussing a technologically relevant topic and have obtained an interesting set of data. The interpretation of their results is done up to a decent level of detail and the conclusions are supported by the data. Therefore, I can recommend publication of the work, however, only after taking into account a list of suggestions and answering a few questions, which should be incorporated in a revised version of the manuscript.

===================

Line 8

metalized > metallized

Line 14

reveals > reveal

Line 19

integrations > integration

Line 41

polymers > polymer

Introduction

Please carefully check for the use of words as ‘single’ or ‘multiple’. I sometimes get the feeling the wrong one of the two is chosen.

Line 45

and therefore, > and, therefore,

Entire paper

The use of [etc] is very high. Please carefully check if that can be reduced to a minimum. I have the impression that at many places this can be removed, as multiple examples are already given.

Line 59

applications > application

Line 73

[like palladium, platinum or gold surrounded by tin ions]

What is the function of tin ions in the process ? That is not explained at all.

Line 79

structures, characterized > structures characterized

Line 93

[well-defined line]

Can this be clarified further ? To this reviewer it is not immediately clear what is meant here.

Line 107

[this experiment]

What experiment are the authors referring to ? Better to write it broader, e.g. [the experiments in this work] , [our experiments] , etc.

Line 132

for injection > for the injection

Line 138

both the selected > both selected

Table 1

Is there a particular reason the authors used 63°C for the etching process step ? It seems a very specific value.

Line 153

details > detail

Line 155

were > was

‘metallization quality’ is the subject of the sentence, therefore is should be single

Line 157

Struers > Struers’

Line 181-185

[Despite the fact that the fingertips are characterized by the lowest 181 surface areas, the copper was able to create there continuous and defect-free coating (Fig 182 3c). This means that the small surface area of ABS is capable to react properly with applied 183 etching and activating solutions to provide surface which is ready for subsequent 184 metallization at the same level as large flat area of the part.]

I don’t understand why SURFACE AREA is an argument here at all. Example: Take a 1 × 1 mm inspection area, both for a surface at a finger and at the hemi-circular part of the injection molded sample. Then when both surfaces get treated, they ‘see’ the same chemical treatments, same electroless deposition process, etc. Why is SURFACE AREA important here at all, then ? If diffusion does not play a significant role in the electroless metal deposition chemistry, Cu deposition rate should be the same at these 2 parts of the sample. This should be clarified further in a revised version of the manuscript, and be rewritten.

Line 186-194

The authors are describing diffusion limitation of the supply of Cu2+ ions to convex and concave areas of the sample. Have the authors experimented with vigorous stirring of the electroless metal deposition solution, e.g. using a magnetic stir bar or a propellor-type stirring rod ? I think nothing regarding the ‘hydrodynamic situation’ is mentioned in the Experimental section, which should be added in a revised version. Whether the solution was stagnant or continuously mixed should be made clear.

Line 207-208

[, something intrinsic with the materials that is making 207 this differences. 208] > [suggests there is some intrinsic issue with the polymer material that is causing the difference.]

Line 241

[catalyzation]

Is that a proper English word ?

catalysis of the surface

or:

catalytic action of the surface

Line 246

Some titles are mixed up here

Line 277

microfetures > microfeatures

Line 283 + 305

broken in many places

Is it really ‘broken’ or is it ‘discontinuous’. ‘Broken’ suggests that it was connected before ‘something’ happened.

Line 302

homogenous > homogeneous

Line 324

blend– between > blend it varies between   

Line 325

results > result

Line 327

parts mean > parts which means

Line 337-340

[For smaller fingers having larger surface to 337 volume ratio, catalyzed palladium sites and subsequent copper particles depositing sites 338 are placed closer to each other and their merger proceeds faster, resulting in more 339 continuous and higher quality metal film.]

Is that really correct ? Does the catalyst density (number of catalyst particles per unit surface area) on the surface DEPEND on the size of the feature on which it is deposited ? How does that work then, mechanistically ? Does this mean that diffusion plays a role in the catalyst deposition process ? Did the authors perform experiments in which they enhance the supply of solution to the surface on which adsorption is needed, for instance by vigorous stirring of the activation solution ?

Line 356

have significant > have a significant

Line 362

shows > show

Line 368

catalization

Is that proper English ?

Line 369

samller > smaller

Line 376

subsequent > the subsequent

Line 388

process > processes

Author Response

(The authors gave the same response as above.)

Round 2

Reviewer 2 Report

The reviewer would like to thank the authors for preparing a revised version of the manuscript. There are still a number of English language mistakes of which I would be expecting the authors to not put them in a revised version and that before resubmitting those would have been captured. Somewhat  disappointing !

Furthermore, I am not fully convinced by a few of the reasonings in the revised version and, therefore, have to reject the paper. See my detailed comments, below: 

========= 

Line 70

groves > grooves

Line 77

can wets

Check if this is proper English, I think it should be ‘can wet’

Line 100

easyly > easily

Line 196-199

[For successful metallization, 196 enough area of surface needs to be provided so that the chemicals used in various 197 metallization steps can interact with the surface and can make the necessary changes of 198 the surface that is needed for metallization.]

What is defining how much is enough ? This is a qualitative story and I don’t see such strong claims being supported by the experimental data.

Line 201

were possible > was possible

Line 302

discontinious > discontinuous

Line 324

discontinious > discontinuous

Line 356 and further

Discussion about ‘surface to volume ratio’. In my opinion, the authors have obtained NO proof whatsoever that the surface-to-volume ratio is REALLY the determining factor. If the authors disagree, then please explain more in detail what proof is obtained and include it in the paper. Isn’t ‘just’ the ‘surface quality’ (in terms of surface roughness, surface chemistry, etc) at play here, that leads to different surface areal densities of the palladium catalyst, which then get ‘translated’ into different amounts / uniformity of Cu metallization afterwards ? If surface-to-volume ratio really plays a role, then FOR SURFACES WITH THE SAME ‘SURFACE QUALITY’, it should matter how much ‘bulk material’ is present ‘behind the surface that is being studied’. How does that work then, mechanistically ? (I would be surprised if the BULK of the material which SURFACE is being metallized does play any role) I don’t see any way to use the argument of surface-to-volume ratio in this particular materials system. Indeed, smaller fingers have a LARGER surface-to-volume ratio, and do show a larger areal density of catalyst particles. However, when there is a correlation it does not necessarily mean there is a causation ! In this case, the authors claim there is a causal relation between surface-to-volume ratio and density of catalyst particles. Please explain the mechanism.

Did the authors perform a detailed study of what I called above the ‘surface quality’ of tips of various sizes, i.e. surface-to-volume ratio ? (for instance using AFM to study the surface roughness, that might lead to different catalyst particle adhesion ?) That might be a way to scientifically investigate the underlying mechanism / cause of the observations. I don’t like the current handwaving argumentation and quaLitative story that linguistically might make sense and read correct, however, I am not convinced it is scientifically sound and supported by experimental results.  

Line 362

catalized

Please check English  

Line 420

injeciton > injection

Author Response

Authors' responses to the comment of the 2nd reviewer are given in the attached document. We thank the reviewer for the valuable comments and suggestions. It really helped to raise the quality of the paper.

Round 3

Reviewer 2 Report

The reviewer would like to thank the authors for submitting a revised version of their work, which has taken into consideration the questions and concerns raised in the previous round. Additional comments placed in the Discussion and Conclusions sections have better clarified the reasoning, indicated some limitations of the current experimental results, as well as a few suggestions for further research. Therefore, the work can be accepted.